# Probing Social Bias in Labor Market Text Generation by ChatGPT: A Masked Language Model Approach

**Lei Ding**[1*]**, Yang Hu**[2]**, Nicole Denier**[3]**, Enze Shi**[1]**, Junxi Zhang**[5]**,**
**Qirui Hu**[6]**, Karen D. Hughes**[3,4]**, Linglong Kong**[1*]**, Bei Jiang**[1*]

[1]Department of Mathematical and Statistical Sciences, University of Alberta, Canada
[2] Department of Sociology, Lancaster University, UK
[3] Department of Sociology, University of Alberta, Canada
[4] Department of Strategy, Entrepreneurship and Management, University of Alberta, Canada
[5] Department of Mathematics and Statistics, Concordia University, Canada
[6] Department of Statistics and Data Science, Tsinghua University, China
`{lding1,nicole.denier,eshi,khughes,lkong,bei1}@ualberta.ca`
`yang.hu@lancaster.ac.uk, junxi.zhang@concordia.ca, hqr20@mails.tsinghua.edu.cn`

## Abstract

As generative large language models (LLMs) such as ChatGPT gain widespread adoption in various domains, their potential to propagate and amplify social biases, particularly in high-stakes areas such as the labor market, has become a pressing concern. AI algorithms are not only widely used in the selection of job applicants, individual job seekers may also make use of generative LLMs to help develop their job application materials. Against this backdrop, this research builds on a novel experimental design to examine social biases within ChatGPT-generated job applications in response to real job advertisements. By simulating the process of job application creation, we examine the language patterns and biases that emerge when the model is prompted with diverse job postings. Notably, we present a novel bias evaluation framework based on Masked Language Models to quantitatively assess social bias based on validated inventories of social cues/words, enabling a systematic analysis of the language used. Our findings show that the increasing adoption of generative AI, not only by employers but also increasingly by individual job seekers, can reinforce and exacerbate gender and social inequalities in the labor market through the use of biased and gendered language.

## 1 Introduction

The rapid advancements in generative Large Language Models (LLM) like ChatGPT [OpenAI, 2023], mark a significant technological shift. These models have not only propelled the field of Natural Language Processing (NLP) but have also found widespread application across numerous sectors [Zhao et al., 2023, Yang et al., 2024]. However, as these models are incorporated into social and economic practices, they bring to the fore critical ethical concerns, especially regarding their potential to propagate and amplify existing social biases and attendant inequalities, particularly within high-stakes domains such as the labor market [Liang et al., 2022].

Recognizing the growing potential of generative AI use in employment practices, our research primarily aims to identify and understand the impact of biases in the application of generative LLM within the labor market. We focus particularly on ChatGPT, investigating how this widely used LLM influences the propagation of biases in job advertising and application processes.

---

[*]Corresponding Authors

38th Conference on Neural Information Processing Systems (NeurIPS 2024).

The complexity of *automating* bias evaluation in textual content poses significant challenges. Traditional approaches in social sciences, such as content analysis, often rely on manual word counts from static lists [Gaucher et al., 2011], which may miss the subtleties and unlisted language cues that advanced NLP technologies can detect. In addition, by considering words individually, these traditional approaches often fail to capture the contextual meanings that emerge from the interplay of words within entire sentences. To address this limitation and build toward a more solid bias evaluation method, we develop a novel bias evaluation algorithm called **PRISM**: **P**robability **R**anking b**I**as **S**core via **M**asked language model. PRISM involves masking words sequentially within texts and using the Masked Language Models (MLM) [Devlin et al., 2018, Liu et al., 2019] to predict the likelihood of alternative tokens, thus allowing us to assess bias with a ranking-based approach that leverages established word lists from social science research to provide contextual sensitivity, enabling a systematic and detailed analysis of language use.

Additionally, the inherently opaque nature of LLMs like ChatGPT, which function as black boxes without transparent access to their internal structures or parameters, adds another layer of complexity. We propose a method of probing these biases by simulating and analyzing how job seekers use ChatGPT to craft applications (output texts) in response to real job postings (input texts), as illustrated in Figure 1. This simulation reveals insights into the biases embedded within ChatGPT's training data and their potential impacts on real-world human resource practices.

Utilizing our PRISM algorithm in tandem with job posting and application text pairs, we explore the correlation between generated content and bias propagation. This comprehensive and novel simulation offers a distinctive lens through which to view how biases might influence the job application process.

In essence, this paper seeks to bridge the gap between rapid technological advancements and the ethical considerations raised by the use of generative LLMs. Through our research, we emphasize the importance of ensuring that AI use promotes core social values of fairness and equality in the labor market as these technologies become increasingly integral to our daily lives.

Our key contributions include:

- We propose **PRISM**, a brand new paradigm for bias evaluation combines with validated word lists capturing directional cues (based on social science research) with MLM to assess biases in texts. It advances existing methods in terms of efficiency, flexibility, robustness as well as theoretical properties.

- We draw on a novel experimental design to probe the black-box of social biases in ChatGPT models to understand both the biases inherent in their training data and their implications for real-world job application scenarios.

- Analysis of bias across four different social dimensions demonstrates inherent biases in job postings are likely reproduced in ChatGPT-generated job applications, with a tendency for the model to exacerbate and reinforce these biases.

This paper is structured as follows: we first review the current landscape of bias evaluation in NLP and social sciences. Following this, we introduce our bias scoring algorithm and provide experimental evidence supporting our methodology. We conclude with an analysis of job postings and applications mediated by ChatGPT, evaluating our approach's broader applicability and discussing the social implications of our empirical findings.

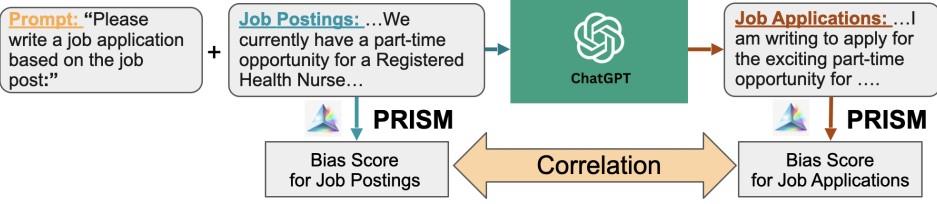

Figure 1: Overview of the paradigm for bias probing experimental design.

## 2  Background and Related Works

**Bias & Fairness Evaluation in NLP**   The evaluation of bias within natural language processing (NLP) presents complex challenges, as methodologies vary significantly across studies [Garrido-Muñoz et al., 2021, Blodgett et al., 2020]. Traditional approaches range from analyzing cosine similarity in word embeddings [Bolukbasi et al., 2016] to diverse methods such as correlation, clustering, classification, and visualization [Caliskan et al., 2017, Gonen and Goldberg, 2019, Ding et al., 2022, Shi et al., 2024]. Recent works have focused on detecting bias in language models that rely on manual sentence templates [Kurita et al., 2019] or creating benchmarks that require high-cost crowd-workers [Nangia et al., 2020, Nadeem et al., 2021] and across various NLP downstream tasks including text classification [De-Arteaga et al., 2019, Blodgett et al., 2016], coreference resolution [Zhao et al., 2018], natural language inference [Dev et al., 2020], and machine translation [Stanovsky et al., 2019].

**Bias Evaluation for Text**   The domain of text bias evaluation is notably more challenging than evaluating the NLP models, often requiring extensive human expert intervention or resorting to simplistic and heuristic methodologies. Many existing approaches are also limited to specific types of bias, making them difficult to adapt to other contexts. Dhamala et al. [2021] measure bias by computing the cosine similarity of word embeddings [Mikolov et al., 2013, Pennington et al., 2014] with respect to the gender direction $(\overrightarrow{he}-\overrightarrow{she})$ [Bolukbasi et al., 2016] and averaging over sentences. Cryan et al. [2020] compare a lexicon-based approach and a fine-tuned BERT model with a Crowdsourced label dataset. Spinde et al. [2021] developed a media bias dataset through costly expert annotation, a process not easily generalizable to other domains. Raza et al. [2024] explore the use of named entity recognition for detecting biased words within texts. Yet this approach also requires the creation of costly labeled training data for each task and model training.

**Labor Market Bias Evaluation in Social Sciences**   A substantial body of research has documented prevalent gender stereotypes and their role in (re)producing inequalities – gender segregation [Kjeldstad and Nymoen, 2012, England, 2010], gender wage/promotion gaps [Blau and Kahn, 2020], motherhood penalties [Glauber, 2018], and fatherhood premiums [Killewald, 2013] – in the labor market. Further research shows that gendered language plays a crucial role in maintaining and reproducing gender stereotypes [González et al., 2019]. Psychological studies also show that women and men, given their gender socialization, tend to use and be attracted to different gendered languages and linguistic styles [Gaucher et al., 2011]. For example, women tend to employ and identify with a more communal language style, including the use of words related to social and emotional contexts [Bem, 1974]. In contrast, masculine language is typically characterized by a style that highlights agentic traits. Gendered language is found across a wide range of contexts, and in the labor market, it features prominently in job advertisements, the language used in job applications and interviews, as well as performance management processes [Hu et al., 2024, 2022a]. While existing research has often focused on gendered language from the labor demand side in terms of, for example, employers' wording of job advertisements [Hu et al., 2022b], far less attention has been paid to the language used by job candidates in response to job advertisements in order to secure a job, despite an increase in individual job seekers' use of ChatGPT. This study thus fills this important gap by assessing gendered languages from both the labor demand (job advertising) and supply (job application) sides. In doing so, it highlights the relational use of language in the job application process as a quintessential example of social interactions in action. It aims to explore and reveal the extent to which gender biases are present and indeed circulated and exacerbated through the interplay between languages used in job advertisements and job applications.

## 3  Bias Evaluation Algorithm for Text

### 3.1  Motivations

When assessing social bias in textual content, previous methodologies often begin with a straightforward approach: selecting keywords for simple frequency counts. For instance, this might involve comparing the total word count of feminine and masculine words. This technique is prevalent in psychological and sociological studies as described in Section 2. More contemporary methods have advanced to include the use of static word embeddings to measure semantic similarities among words,

although these approaches still treat each word individually. To go further, researchers need to acquire expensive, labeled training data for specific tasks and do the model training.

In contrast, our objective is to refine and further advance these existing approaches to measuring textual bias with three useful and practical settings:

- Beyond merely analyzing each word individually, the algorithm should aim to consider the contextual meanings of entire sentences, allowing for a more nuanced and comprehensive view of the text.

- The algorithm does not require costly human-labeled training data and circumvents the process of model training or fine-tuning. This aspect is particularly valuable in scenarios where the necessary labeled data is not readily available, allowing for more flexible and scalable applications.

- The algorithm should incorporate established and rigorous word inventories from social science research to guide the bias calculation in a contextually embedded and domain-specific manner (e.g., accounting for specificities of the labor market context). This incorporation of domain knowledge ensures that the assessments are both empirically grounded and contextually salient.

## 3.2    Problem Setup and Algorithm Implementation

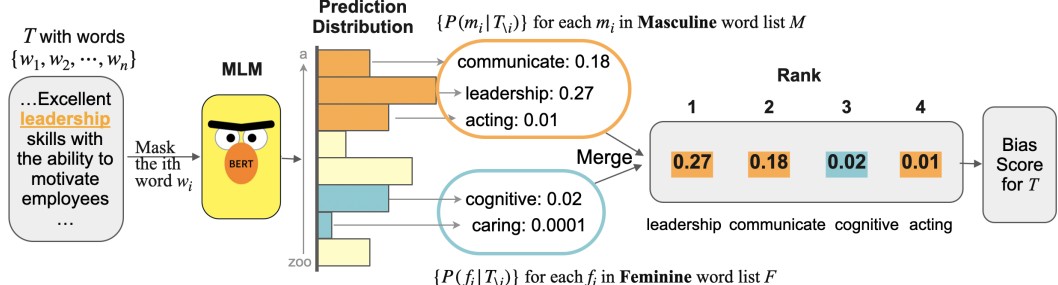

Figure 2: An illustration of the paradigm for **PRISM** that uses word lists for directional cues with MLM to compute bias score for text.

In this section, we detail our algorithm under the settings introduced above. Given a text $T$ comprising $n$ words $T = \{w_1, w_2, \ldots, w_n\}$, we iteratively mask each word $w_i$ and input the modified masked text $T_{\backslash i} = \{\ldots, w_{i-1}, [\text{MASK}], w_{i+1}, \ldots\}$ into an MLM, which outputs the probability distribution over the vocabulary for the masked position $i$, denoted as $P(\cdot \mid T_{\backslash i})$.

Then, to obtain the direction signal for score calculation, we require two predefined word lists representing different contexts—such as gender with a feminine word list $F = \{f_1, \ldots, f_{|F|}\}$ and a masculine word list $M = \{m_1, \ldots, m_{|M|}\}$. For each word in $F$ and $M$, we obtain the probability from the distribution $P(\cdot|T_{\backslash i})$. This yields two sets of probabilities: $P_F = \{P(f_1|T_{\backslash i}), P(f_2|T_{\backslash i}), \ldots, P(f_{|F|}|T_{\backslash i})\}$ and $P_M = \{P(m_1|T_{\backslash i}), P(m_2|T_{\backslash i}), \ldots, P(m_{|M|}|T_{\backslash i})\}$.

Next, we merge $P_F$ and $P_M$ and filter the probabilities by taking the top $\alpha$ percent of the probabilities, as lower probabilities represent less likely predictions by the MLM and thus contribute minimally to our analysis of bias. This step allows us to focus on the most influential predictions which significantly determine the context of the sentence.

Finally, we calculate the rank of each probability within this merged list. Let $R(\cdot)$ denote the rank function of the probability based on the merged list. For each word $w_i$ in the text $T$ using the word lists $F$ and $M$ is denoted as $R(P(f|T_{\backslash i}))$ and $R(P(m|T_{\backslash i}))$, respectively. And the lower the rank indicates the higher the probability. The bias score for each word $w_i$ is computed as the difference between the mean ranks of the two word lists:

$$S(w_i) = \frac{1}{|F|} \sum_{f \in F} R(P(f|T_{\backslash i})) - \frac{1}{|M|} \sum_{m \in M} R(P(m|T_{\backslash i}))$$

A positive score indicates a bias toward a masculine orientation, while a negative score suggests a bias toward a feminine orientation. This differential allows us to detect the direction of the bias, providing deeper insights into how gender nuances are embedded within the language.

Finally, the overall bias score for the text $T$ is the mean of the scores for all words in the text:

$$B(T) = \frac{1}{n} \sum_{i=1}^{n} S(w_i)$$

This score quantifies the bias present in $T$. By analyzing these scores across various texts, we can evaluate both the extent and direction of linguistic bias, offering insights into the underlying gender biases conveyed through language. From a Bayesian perspective, in the absence of prior information, the mean serves as a robust estimator of central tendency. For future work, we recognize the potential benefit of incorporating additional information (e.g., part-of-speech tags, WordNet, etc.) to apply a weighted average or explore alternative aggregation methods, which could further enhance the performance of the bias assessment. The overall algorithm is illustrated in Figure 2, and is detailed in Algorithm 1.

---

**Algorithm 1** PRISM: Probability Ranking bIas Score via Masked language model

---

**Input:** Text $T$ with $n$ words $\{w_1, w_2, \ldots, w_n\}$, Feminine word list $F$, Masculine word list $M$
**Ouput:** Bias score $B(T)$

1: **for** each word $w_i$ in $T$ **do**
2:     Create $T_{\backslash i}$ by masking $w_i$ in $T$
3:     Predict distribution $P(\cdot \mid T_{\backslash i})$ using MLM
4:     Initialize $P_{merged} = [\,]$
5:     **for all** words $w$ in $F \cup M$ **do**
6:         Append $\{w, P(w|T_{\backslash i})\}$ to $P_{merged}$
7:     **end for**
8:     Sort and filter $P_{merged}$ to retain top $\alpha\%$ of entries
9:     Calculate ranks for $R(P(f|T_{\backslash i}))$ and $R(P(m|T_{\backslash i}))$ in the filtered list
10:     $S(w_i) = \frac{1}{|F|} \sum_{f \in F} R(P(f|T_{\backslash i})) - \frac{1}{|M|} \sum_{m \in M} R(P(m|T_{\backslash i}))$
11: **end for**
12: $B(T) = \frac{1}{n} \sum_{i=1}^{n} S(w_i)$
13: **return** $B(T)$

---

### 3.3 Methodological Benefits of PRISM

**Efficiency**   Our algorithm eliminates the need for costly data labeling and model training. By leveraging predefined word lists developed by existing sociological research, our method avoids the resource-intensive processes associated with supervised learning, such as gathering expert annotations and training models from scratch. This approach not only expedites deployment but also ensures that the algorithm can be scaled and adapted swiftly and economically, making it highly practical for researchers and practitioners needing quick and reliable bias assessments in various settings.

**Computational Flexibility**   The inherent flexibility of our method allows for the evaluation of bias across various dimensions simply by altering the word list cues. This adaptability means that different types of bias can be assessed without the need to relabel data or retrain models, significantly reducing the time and resources required for analysis. Whether exploring gender, race, age, or any other form of bias, our algorithm can adjust to new research questions with minimal adjustments. This also allows for the incorporation of substantively meaningful domain-specific word inventories from social science disciplines such as sociology, management studies, psychology, etc.

**Robustness**   Robustness in our method is two-fold. Firstly, we utilize ordinal measurements of word probabilities, focusing on relative positions (ranking) rather than the values. This method effectively mitigates issues arising from the predominance of low probabilities within a large pool of candidate words, which can lead to nonsensical outcomes. Secondly, our approach ensures robust

results across different MLMs. Unlike other scoring methods using raw probabilities for calculation, our rank-based bias score method remains consistent even when different MLMs produce varying output probabilities. This dual approach minimizes the influence of outliers and maintains reliability across various computational models.

**Theoretical Properties**   Moreover, we can test whether MLM's predictions have the same distribution on two word lists ($M$ and $F$). Consider two groups of probabilities scores, $\{P(f|T_{\setminus i})\}_{f \in F}$ and $\{P(m|T_{\setminus i})\}_{m \in M}$, representing the scores samples from distribution $\mathcal{P}_F$ and $\mathcal{P}_M$. The rank sums, denoted by $\sum_{f \in F} R(P(f|T_{\setminus i}))$ and $\sum_{m \in M} R(P(m|T_{\setminus i}))$ respectively, allow us to test the hypotheses $H_{i0} : \mathcal{P}_F = \mathcal{P}_M$ versus $H_{i1} : \mathcal{P}_F \neq \mathcal{P}_M$. The null hypothesis holds if there is no statistically significant bias toward masculine or feminine language in a particular word $w_i$. The following theorem provides a rigorous formulation of the test statistic and its asymptotic result.

**Theorem 1** *When $|F|$ and $|M|$ are large, for each $i \in [n]$, under $H_{i0}$:*

$$\sum_{m \in M} R(P(m|T_{\setminus i})) \sim N \left( \frac{|M|(|F| + |M| + 1)}{2}, \frac{|F||M|(|F| + |M| + 1)}{12} \right)$$

*If further we have $|M| = |F| = K$, for each $i \in [n]$, under $H_{i0}$:*

$$S(w_i) \sim N \left( 0, \frac{2K + 1}{3} \right)$$

Note that we interpret the prediction probabilities for the two word lists, $\{P(f|T_{\setminus i})\}_{f \in F}$ and $\{P(m|T_{\setminus i})\}_{m \in M}$, as scores that measure the association between the masked word and the words in the lists. A higher score indicates a stronger relationship. We also assume these word lists are sampled from larger sets of male or female-associated words. Thus, for each masked word, the scores $\{P(f|T_{\setminus i})\}_{f \in F}$ and $\{P(m|T_{\setminus i})\}_{m \in M}$ are viewed as samples from two underlying distributions, $\mathcal{P}_F$ and $\mathcal{P}_M$. Using the rank test from Theorem 1, we can determine whether $\mathcal{P}_F = \mathcal{P}_M$. This test is particularly useful for detecting biased words or sentences. For non-biased words, the associations with the two word lists should be similar, implying $\mathcal{P}_F = \mathcal{P}_M$. However, for biased words, the associations differ significantly, resulting in $\mathcal{P}_F \neq \mathcal{P}_M$.

### 3.4   Algorithm Validation

To demonstrate the reliability of our scoring algorithm in identifying social biases within texts, we validate our method on two different tasks[2]:

**Human Experts Validation**   This validation involved collaboration with six experienced professionals from the fields of sociology and management science. Each coder manually labeled a randomly selected subsample of job advertisements. Leveraging their extensive domain knowledge, these experts meticulously classified the advertisements, assessing them for levels of perceived gender bias. These categorical labels were then transformed into ordinal variables, enabling a detailed statistical comparison with the results produced by our scoring algorithm. This rigorous, expert-driven coding process ensured the reliability of our evaluation methodology.

We compute the Spearman rank correlation between the bias scores generated by our algorithm and the results from the manual labeling process. A Spearman correlation coefficient of 0.85[3] was obtained (Figure 4a), indicating a strong positive association between our algorithm's scores and the human experts' assessments. This result validates the algorithm's capacity to accurately reflect human judgments of bias, confirming its effectiveness as a tool for social bias detection.

**Benchmark Validation**   Further validation was conducted using the BIOS dataset [De-Arteaga et al., 2019], which comprises personal biographies categorized by gender and various occupations. We employed gender-specific word lists from [Konnikov et al., 2022], such as {man, his, he ...} versus {woman, her, she...}, as binary directional cues and designated gender as the ground truth label. Our algorithm demonstrated high performance, achieving an AUC of 0.97 in classifying gender, as

---

[2]The code for the algorithm is available at: `https://github.com/Lei-Ding07/ChatGPT_bias/`

[3]This correlation is notably higher compared to those typically observed in non-experimental social sciences.

illustrated in Figure 4b. The AUC, or Area Under the ROC Curve, measures the ability of our model to distinguish between classes — here, gender categories. This performance surpasses that of three baseline methods in [Dhamala et al., 2021] that rely on unigram or word embeddings, highlighting the effectiveness and potential applicability of our bias detection approach in broader NLP tasks.

# 4 Probing Methodology and Job Application Data Generation

**Probing Methodology** To explore the social biases inherent in ChatGPT, particularly in the context of the labor market, our study simulates the typical use case where job seekers employ ChatGPT to assist in drafting job applications. This approach allows us to investigate not only the biases that may emanate from ChatGPT's training data but also to understand how these biases could potentially influence real-world job application/hiring processes.

Probing the social biases within ChatGPT presents several challenges. Firstly, ChatGPT's model operates as a 'black box,' making it difficult to discern the internal processes that contribute to bias propagation. Secondly, the lack of access to the model's architecture or parameters further complicates direct examination. Therefore, our analysis adopts an indirect method, employing our known bias evaluation algorithm to detect and quantify the biases exhibited by ChatGPT, thereby illuminating how these biases might manifest in practical applications.

**Job Application Data Generation** Our dataset comprises over 33K job postings collected from LinkedIn, reflecting a diverse range of industries and job types. To simulate realistic job application processes, we utilize the OpenAI API (GPT-3.5 Turbo, data collected on April 2024) to prompt ChatGPT with these job advertisements, instructing it to generate corresponding job applications for each job posting.

This method does more than replicate real-world scenarios where individuals respond to job postings—it also facilitates a comprehensive analysis of the generated texts across various sectors. By using job advertisements as standardized prompts, we ensure that any observed deviations from neutrality in the generated texts are attributable to the model's ingrained biases, rather than the content of the advertisements themselves. This setup is crucial for isolating the effects of ChatGPT's biases, allowing for an accurate assessment of bias presence and intensity using the quantifiable metrics provided by our bias score calculation method.

# 5 Analysing the Bias inside ChatGPT

## 5.1 Dimensions of Gender Bias

We begin by introducing the four gender dimensions, each defined by a distinct set of gender-related word lists, which will form the basis of our analysis. In recent social science research, understanding gender bias involves not just recognizing the existence of biases but also evaluating their impacts in various contexts. Building on the framework proposed by Bem [1974], Gaucher et al. [2011], Konnikov et al. [2022], we utilize specialized word lists to apply our social bias analysis across four different dimensions. Each dimension not only helps identify specific instances of bias but also offers insights into the broader social and psychological dynamics at play.

**Psychological Cues:** The psychological dimension assesses language context leaning towards communal attributes (e.g., "caring," "sympathetic," "attentive") commonly associated with femininity, or agentic attributes (e.g., "authoritative," "active," "confident") typically linked to masculinity.

**Role Description:** We evaluate job descriptions and roles using word lists that categorize terms associated with "soft" and "social" skills for feminine orientation, and "time-compressed" and "stressful" tasks, such as "multitasking," "pressure," "speed," for masculine orientation.

**Work–Family Characteristics(WFC):** This dimension examines employer policies and cultural expectations affecting gendered labor force participation, scrutinizing terms like "parental leave" and "flexible work" for feminine orientation versus "irregular and long work hours" and "weekend work" for masculine orientation.

**Social Characteristics:** We also analyze explicit gender references such as gendered pronouns and identity markers ("she," "he," "his," "her," "man").

## 5.2 Correlation Analysis

We first analyze the correlation of job postings and job applications across each dimension of gender bias. Our findings indicate a consistent positive linear correlation between the bias scores of job postings and the ChatGPT-generated job applications. This trend suggests that the biases inherent in job postings are likely to be reproduced in job applications by generative AI, reinforcing and possibly amplifying the initial biases. This correlation is visually captured in Figure 3, illustrating the potential for cyclical reinforcement of biases through the use of generative AI in job application practices.

Figure 3 presents the statistical parameters for each analyzed dimension of social bias. The strongest correlation is observed in the Social Characteristics dimension with a correlation coefficient of 0.777, indicating a very strong positive relationship. This is followed by the Role Description dimension, which shows a correlation coefficient of 0.708. Both of these correlations suggest significant potential for the biases in job postings to be reproduced by AI in job applications in these dimensions.

The Psychological Cues and WFC dimensions exhibit lower but still substantial correlation coefficients of 0.644 and 0.451, respectively. The slopes of these relationships indicate the rate at which the bias scores from job postings predict those in job applications, with steeper slopes observed in the Social Characteristics dimension. This analysis clearly supports the hypothesis that inherent biases in job postings are likely reproduced in ChatGPT-generated job applications.

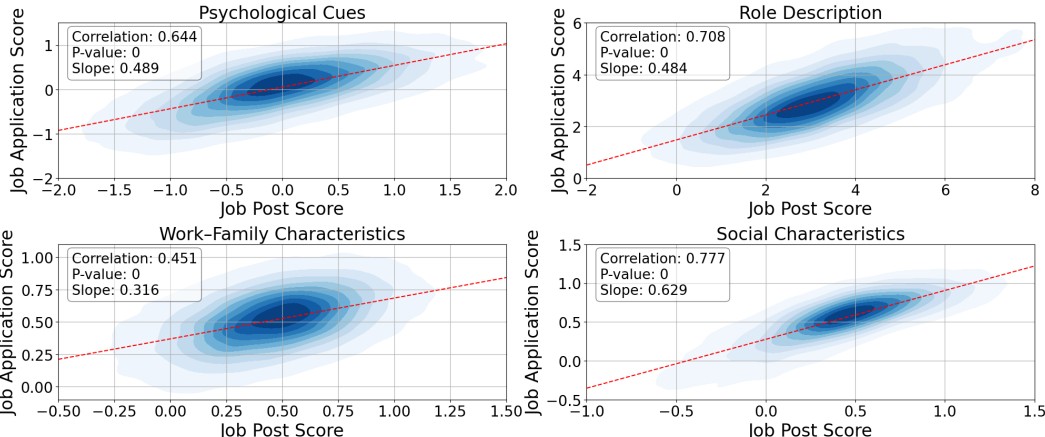

Figure 3: Result scatter density plot, for each of the bias dimensions where the x-axis is the job posting bias score and the y-axis is the job applications bias score. Where the darker color means there are more dots. The p-value is the significance of the correlation coefficient.

## 5.3 Statistical Testing Analysis

In this section, we delve deeper into how ChatGPT influences bias reproduction within the job application process. Let $X$ and $Y$, represent the bias scores of job postings and those of the job applications generated by ChatGPT, respectively. We denote the population mean and variance of $X$ as $\mu_X$ and $\sigma_X^2$. A score close to zero indicates minimal bias (i.e., gender neutrality that is neither feminine nor masculine), a higher positive score signifies a bias towards masculine language, and a lower negative score indicates a bias towards feminine language. The aim is to evaluate how ChatGPT may exacerbate or mitigate these biases. The histogram and summary statistics of the bias scores are in Appendix A.5.

**Shift in Mean**  We propose the following hypothesis tests to assess shifts in mean:

$$H_0 : \mu_X \geq \mu_Y \quad \text{vs.} \quad H_1 : \mu_X < \mu_Y$$

Using the Wilcoxon signed-rank test, we determine whether there is a significant change in the mean bias score from the job postings to the applications.

**Shift in Magnitude**  For the magnitude of bias, we assess:

$$H_0 : |\mu_X| \geq |\mu_Y| \quad \text{vs.} \quad H_1 : |\mu_X| < |\mu_Y|$$

This test measures the central tendency of bias scores, examining if the absolute values (regardless of bias direction) decrease. The less the magnitude(i.e. closer to zero) the less bias it has.

**Change in Variance**    We also explore the variability in bias scores:

$$H_0 : \sigma_X^2 \leq \sigma_Y^2 \quad \text{vs.} \quad H_1 : \sigma_X^2 > \sigma_Y^2$$

This variance test, employing Levene's test [Brown and Forsythe, 1974] for equality of variances, explores whether ChatGPT produces job applications with more uniform bias expressions compared to the job postings. It helps determine if there is a reduction in variance, which would suggest that ChatGPT standardizes the use of gendered language cues. Such standardization could potentially reinforce specific gender biases more consistently.

Table 1: Statistical testing results for each dimension. The mean result indicates whether the overall bias score is shifting toward the masculine ($\uparrow$) or feminine ($\downarrow$) direction. The magnitude result reveals whether the bias is moving toward zero ($\downarrow$) or away from zero ($\uparrow$). The variance assesses whether job application bias scores exhibit greater ($\uparrow$) or lesser ($\downarrow$) variance compared to the job postings. Please refer to Table 2, 3, 4 and 5 in Appendix for detail statistics.

| Dimensions | Mean | Magnitude | Variance |
|---|---|---|---|
| Psychological Cues | $\uparrow$ | $\downarrow$ | $\downarrow$ |
| Role Description | $\downarrow$ | $\downarrow$ | $\downarrow$ |
| Work–Family Characteristics | $\uparrow$ | $\uparrow$ | $\downarrow$ |
| Social Characteristics | $\uparrow$ | $\uparrow$ | $\downarrow$ |

**Shift in Mean**    The testing for the mean shift in Table 1 reveals significant findings across several dimensions. Except for Role Description, all other dimensions exhibit statistically significant shifts toward more masculine language. This indicates a predominant inclination for ChatGPT to amplify the use of masculine language in simulated job applications over and above the original job postings, possibly due to its training on historically biased data. This shift raises concerns about the consolidation and exacerbation of masculine language. Such biases in AI-generated content could perpetuate gender disparities in professional settings, emphasizing the need for interventions in AI training processes to address and correct historical biases. In contrast, the Role Description dimension shows a mean shift toward a less masculine direction, but the bias in job postings has already been shown to be skewed toward a very masculine direction. In this case, ChatGPT seems to help mitigate this extreme masculine bias.

**Magnitude of Bias**    The magnitude of bias, assessed through the mean of the absolute bias scores, varies across the dimensions. The Psychological Cues and Role Description dimensions suggest that the overall intensity of bias—regardless of direction—does not increase. This could imply that while the direction of bias towards masculinity is pronounced, the degree of bias embedded within job applications does not intensify. Conversely, the WFC and Social Characteristics dimensions exhibit an increase in bias magnitude, indicating not only a shift towards masculine language but also an overall increase in the strength of biased expressions. This finding is particularly troubling as it suggests that AI-generated job applications in these areas may become more polarized, further entrenching gender-specific expectations in roles traditionally associated with work-life balance and social interactions.

**Variability in Bias Expression**    The variance results across all dimensions reveal a consistent decrease in job applications compared to job postings. This decrease in variance suggests that the language used by ChatGPT is more uniform across different applications, potentially indicating a standardization of language that leans towards masculine expressions. Such uniformity in language use could narrow the range of expressions and perspectives presented in job applications, limiting diversity and potentially skewing hiring decisions in favor of male candidates.

## 5.4    Implications and Extended Analysis

Our statistical results underscore a critical issue: biases in job postings are not merely replicated but are amplified in job applications created by generative AI in response to the postings. This

phenomenon can be explained by the reinforcement of initial biases through the language processing and text generation capabilities of AI tools like ChatGPT, which tend to replicate and often intensify the language patterns they are trained on.

**Societal and Labor Market Implications:** The amplification of gender biases in AI-generated job applications has profound societal and labor market implications, suggesting that not only are stereo-typical roles perpetuated through biased language, but they are also strengthened when individuals use AI tools like ChatGPT to assist with drafting job applications. This use of generative AI plays a crucial role in circulating and amplifying biases, which reinforces, rather than challenges, the gender biases underpinning persistent gender inequalities in the workplace. Such biases can compound, influencing job satisfaction, employee retention, and career advancement. The misallocation of human resources due to biased AI could reduce economic efficiency and innovation, potentially causing sectors to overlook qualified candidates. Furthermore, these persistent inequalities may spur regulatory and legal challenges, especially in countries with robust equal employment opportunity laws, with significant implications for social ethics, justice, and economic equality.

**Recommendations for Intervention:** To mitigate the reproduction of gender biases through LLMs, it is recommended that employers and AI developers implement more rigorous bias monitoring and mitigation strategies. This could include the use of debiased language models, regular audits of AI-generated content by independent third-party organizations, and the development of enhanced AI training datasets that reflect the diversity of the global job market. Additionally, public awareness and education initiatives should be promoted to increase understanding of AI's role in job application and its potential impacts, fostering a critical approach to AI tool usage in professional settings.

## 6   Conclusion

Our paper – including a novel experiment, new algorithm development, and empirical application and findings – contributes to the ongoing debates and developments in the ethical use of AI in labor market processes and practices. By identifying underlying biases in AI-driven text generation, this paper proposes novel strategies and methods for detecting and mitigating such biases. Through our **PRISM** algorithm and empirical application, we show that these strategies are not just theoretical but are intended as actionable steps toward ensuring that the integration of AI in the labor market supports equitable and fair employment opportunities for both employers and job seekers.

## Acknowledgements

This work was supported by the Economic and Social Research Council (ESRC ES/T012382/1) and the Social Sciences and Humanities Research Council (SSHRC 2003-2019-0003) under the scheme of the Canada-UK Artificial Intelligence Initiative. The project title is BIAS: Responsible AI for Labour Market Equality and the Principal Investigator in Canada is Linglong Kong. Bei Jiang and Linglong Kong were partially supported by grants from the Canada CIFAR AI Chairs program, the Alberta Machine Intelligence Institute (AMII), and Natural Sciences and Engineering Council of Canada (NSERC), and Linglong Kong was also partially supported by grants from the Canada Research Chair program from NSERC. Qirui Hu was supported partially by the National Natural Science Foundation of China award 12171269. We thank Haizhou Yu, Qingfeng Lan, and Weitao Zhou for their valuable feedback on this paper. We also thank all the constructive suggestions and comments from the reviewers.

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

## A   Appendix & Supplemental Material

### A.1   Additional Motivation and Real World Impact

The use of LLMs like ChatGPT to generate job applications is increasingly common among job seekers. This trend has been highlighted in discussions and concerns raised by employers and HR professionals, as reported by high-profile news such as CNBC (May 6, 2024, article: "Exact same cover letters word for word: Career consultant says Gen Z are misusing AI") and Vox (Mar 8, 2023, article: "Maybe AI can finally kill the cover letter: Jobs still require cover letters. Apps like ChatGPT can help."). Moreover, the academic community is beginning to explore this use case, focusing both on its technical applications and ethical/bias implications. Our study contributes to this burgeoning body of literature by addressing key public, organizational, and scholarly concerns regarding the use of LLMs in generating job applications.

We explicitly identify the two major harms resulting from bias exacerbation in LLM-generated job applications:

**Structural Harm:** LLM-generated job applications that perpetuate gender stereotypes contribute to the reinforcement of gender inequalities embedded in language used in labor market processes. Job postings and applications are critical steps in these processes and play a significant role in the reproduction of gender inequalities. By failing to challenge these biases, LLMs can inadvertently support the perpetuation of these inequities.

**Practical Harm:** As gender-biased LLM-generated job applications become part of the training data for future AI applications in HR(both in generating job postings and assessing applications), there is a risk of further entrenching gender biases in language use. This entrenchment can lead to cascading effects, such as increased labor force gender segregation, which have significant societal implications.

### A.2   Proof of Theorem 1

We assume each word from two word lists $M$ and $F$ are selected independently. Therefore, the first result in Theorem 1 is implied directly from the Wilcoxon rank sum test [Wilcoxon, 1945].

If $|M| = |F| = K$, the bias score $S(\omega_i)$ can be rewritten as

$$
\begin{aligned}
S(\omega_i) &= \frac{1}{K} \sum_{f \in F} R(P(f|T_{\backslash i})) - \frac{1}{K} \sum_{m \in M} R(P(m|T_{\backslash i})) \\
&= \frac{1}{K} \left[ \sum_{f \in F} R(P(f|T_{\backslash i})) - \left( \frac{2K(2K+1)}{2} - \sum_{f \in F} R(P(f|T_{\backslash i})) \right) \right] \\
&= \frac{2 \sum_{f \in F} R(P(f|T_{\backslash i}))}{K} - (2K+1),
\end{aligned}
\tag{1}
$$

where the second equality follows from the fact:

$$\sum_{f \in F} R(P(f|T_{\backslash i})) + \sum_{m \in M} R(P(m|T_{\backslash i})) = \frac{2K(2K+1)}{2}.$$

From the first result in Theorem 1, we have

$$\sum_{f \in F} R(P(f|T_{\backslash i})) \sim N\left(\frac{K(2K+1)}{2}, \frac{K^2(2K+1)}{12}\right)$$

for each $i \in [n]$, under $H_{i0}$. Therefore, under $H_{i0}$, the result $S(w_i) \sim N\left(0, \frac{2K+1}{3}\right)$ follows directly from the relationship (1) and the distribution of $\sum_{f \in F} R(P(f|T_{\backslash i}))$.

## A.3  Algorithm Validation Result

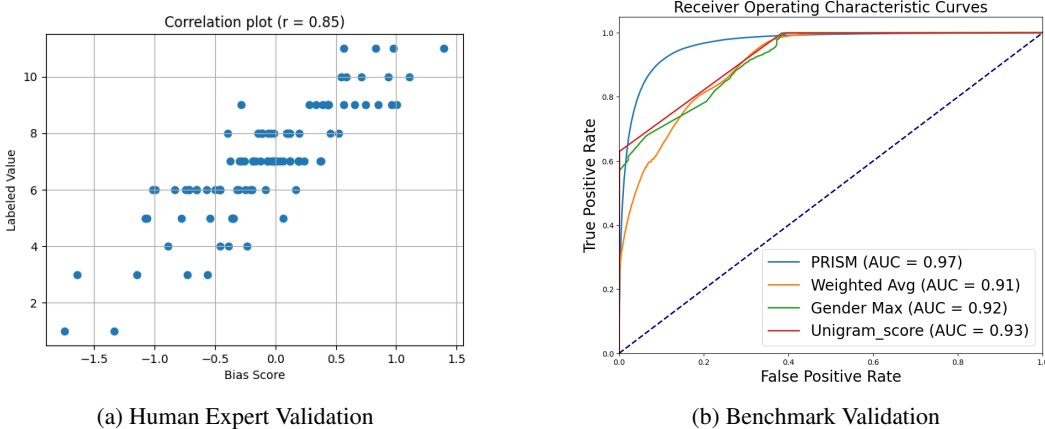

(a) Human Expert Validation

(b) Benchmark Validation

Figure 4: Algorithm Validation

In Figure 4a, A labeled value of 7 signifies neutrality, while values less than 7 suggest femininity, and values greater than 7 imply masculinity. A Bias Score close to zero indicates neutrality, a positive value suggests masculinity and a negative value denotes femininity.

In Figure 4b, We follow three gender metrics for evaluating gender bias in texts in [Dhamala et al., 2021]. The first metric, *unigram matching*, counts gender-specific tokens like 'he', 'him', 'she', 'her' etc., and labels texts with more male tokens as male, more female tokens as female, and texts with equal counts as neutral. The second metric assesses words indirectly related to gender via a normalized projection of word vectors in the gender direction, defined by $\vec{she} - \vec{he}$, using a Word2Vec embedding. Word-level gender scores are calculated as $b_i = \frac{\vec{w}_i \cdot \vec{g}}{\|\vec{w}_i\|\|\vec{g}\|}$. These are aggregated either by a weighted average (*Gender-Wavg*):

$$\text{Gender-Wavg} = \frac{\sum_{i=1}^n \text{sgn}(b_i)b_i^2}{\sum_{i=1}^n |b_i|}$$

or by taking the score from the most gender-polar word (*Gender-Max*):

$$i^* = \arg\max_i(|b_i|), \quad \text{Gender-Max} = \text{sgn}(b_i^*)|b_i^*|$$

Texts are classified as male if the score is less than -0.25 and as female if the score is greater than 0.25.

## A.4  Human Expert Labeling Detail

To ensure the scientific rigor of the evaluation, we paid particular attention to inter-rater validity and reliability. Specifically, each phase included individual labeling of data conducted by four independent

experts specializing in labor market inequalities associated with gender, work, and family, and Equity, Diversity, and Inclusion (EDI) in the labor market. In each round of labeling, individual labeling was followed by group sharing and discussion of the preliminary outcomes among the four experts. This combination of individual and group analysis allowed the team to trace the score labeling and the validation of word inventory, ensure inter-rater reliability in each phase, and contextualize each phase within relevant scholarly literature, policies, and definitions.

The score labeling and the validation were further validated by three additional expert labelers (management, human resource, and social science scholars) from the team. Through a double-blind labeling approach, the three additional experts independently assessed the dimensions and labels produced by the first four experts, demonstrating a high level of consistency. The scores and word lists were then finalized through further deliberation among the four experts and the three additional validators.

Because we used an iterative multi-round coding process, the inter-coder consistency rate in the developmental coding varied between 0.6 and 0.8. Notably, the final validation by three fresh validators within the team achieved a high level of inter-coder consistency exceeding 0.8.

## A.5 Histogram of Bias Scores

In Figure 5, we present the histogram of bias scores for Job Postings and Job Applications on different dimensions.

## A.6 Statistical Tests Results

Below we list the statistical test results for job postings and job application categories across different dimensions, with additional magnitude values, in Table 2, 3, 4, 5.

Table 2: Mean and Standard Deviation for job postings and job application categories across different dimensions, with additional magnitude values.

| Dimension | Job Postings | | | Job Applications | | |
|---|---|---|---|---|---|---|
| | Mean | Magnitude | Std | Mean | Magnitude | Std |
| Psychological Cues | -0.030 | 0.552 | 0.703 | 0.039 | 0.427 | 0.533 |
| Role Description | 3.188 | 3.205 | 1.550 | 3.020 | 3.020 | 1.060 |
| Work–Family Characteristics | 0.455 | 0.476 | 0.290 | 0.513 | 0.515 | 0.203 |
| Social Characteristics | 0.435 | 0.484 | 0.355 | 0.550 | 0.569 | 0.288 |

Table 3: Wilcoxon Test Results for Mean Shift

| Dimension | Statistic | P-value | $H_1$ |
|---|---|---|---|
| Psychological Cues | 229792649.0 | $1.06 \times 10^{-136}$ | $\mu_X < \mu_Y$ |
| Role Description | 316042300.0 | 0.0 | $\mu_X > \mu_Y$ |
| Work–Family Characteristics | 207756451.0 | 0.0 | $\mu_X < \mu_Y$ |
| Social Characteristics | 119987669.0 | 0.0 | $\mu_X < \mu_Y$ |

Table 4: Wilcoxon Test Results on Absolute Values

| Dimension | Statistic | P-value | $H_1$ |
|---|---|---|---|
| Psychological Cues | 353433005.0 | 0.0 | $|\mu_X| > |\mu_Y|$ |
| Role Description | 319570996.0 | 0.0 | $|\mu_X| > |\mu_Y|$ |
| Work-Family Characteristics | 219288122.0 | $8.19 \times 10^{-213}$ | $|\mu_X| < |\mu_Y|$ |
| Social Characteristics | 146778531.0 | 0.0 | $|\mu_X| < |\mu_Y|$ |

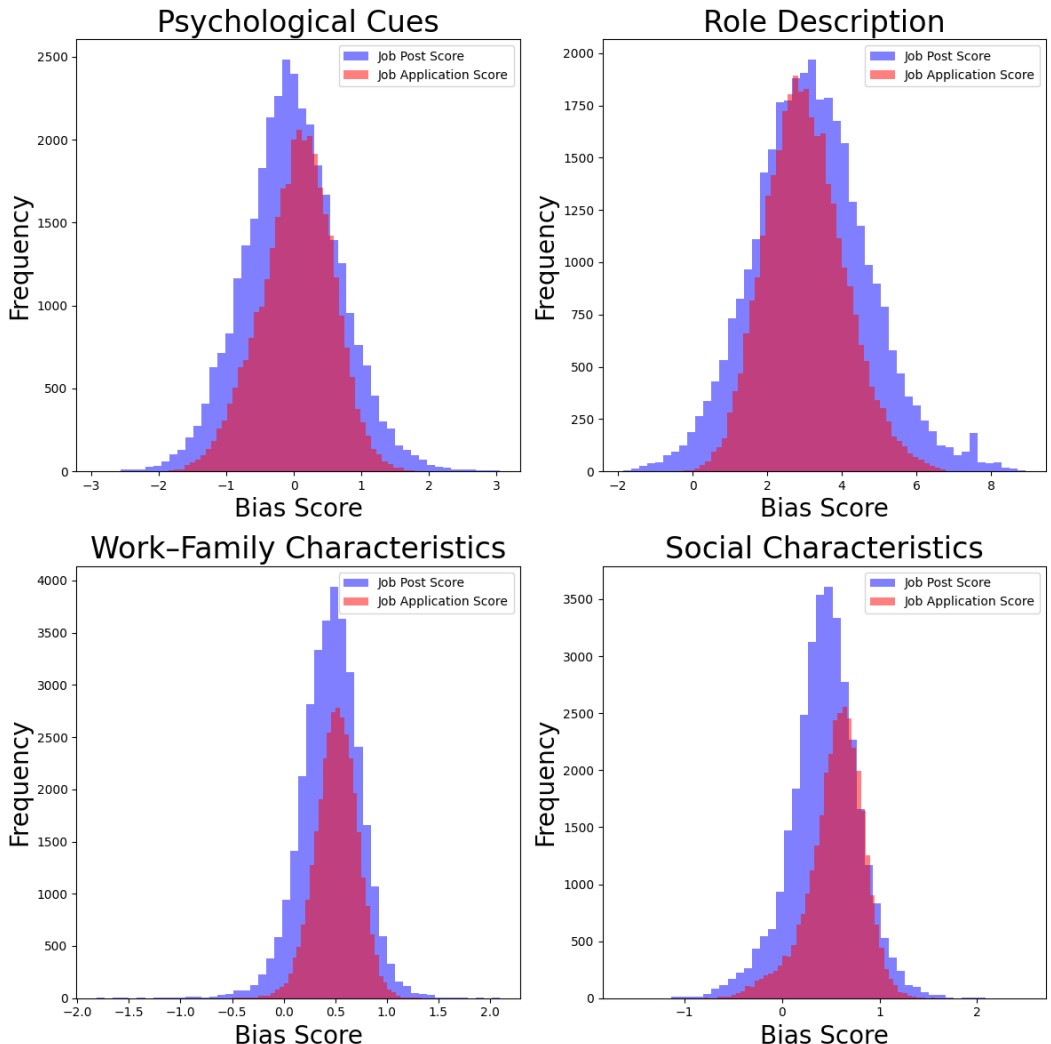

Figure 5: Result histogram, for each of the bias dimensions, we use different colors to distinguish the Job Postings and Job Applications

Table 5: Levene's test results for variance between job and application data across different dimensions, analyzed with a one-sided interpretation. These results indicate significant differences in variance, with the job postings consistently showing greater variance compared to the job applications.

| Dimension | Statistic | P-Value | $H_1$ |
|---|---|---|---|
| Psychological Cues | 1825.094 | 0.0 | $\sigma_X^2 > \sigma_Y^2$ |
| Role Description | 3410.619 | 0.0 | $\sigma_X^2 > \sigma_Y^2$ |
| Work–Family Characteristics | 2491.084 | 0.0 | $\sigma_X^2 > \sigma_Y^2$ |
| Social Characteristics | 922.186 | $1.80 \times 10^{-201}$ | $\sigma_X^2 > \sigma_Y^2$ |

### A.7    Ablation study

Additionally, we conducted Control Experiments with Different MLMs, we selected two more MLMs: BERT-large and DistilBERT. The results for both the human validation and the LLM correlation plots are in Figure 6, 7. These results show that changing the MLMs produces consistent outcomes in the scatter density plot for evaluating bias for LLM, and statistical testing results are also highly consistent.

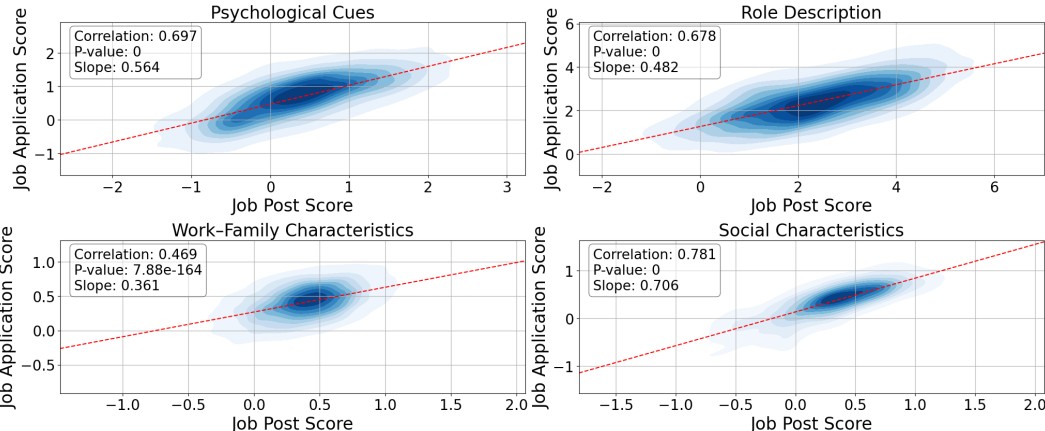

Figure 6: Result scatter density plot for **DistilBERT**

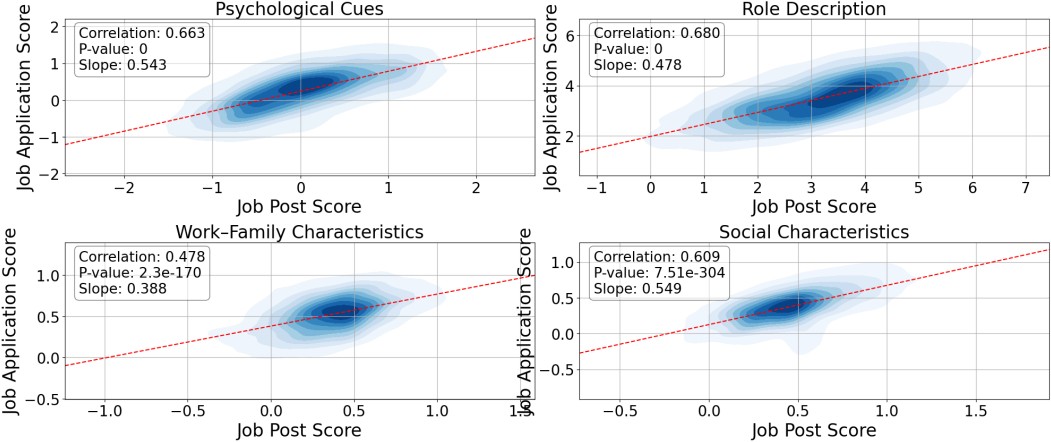

Figure 7: Result scatter density plot for **BERT Based Large**

### A.8    Experiment Setting & Computational Resources

For our analysis, we utilize the 'bert-base-uncased' model from the Hugging Face library. The temperature parameter for the ChatGPT API is set to its default value of 7. For the preprocessing, when we iterate through the text, we skip some of the function words like Articles ("a", "an", "the"), Prepositions ("on", "by"), Conjunctions("and", "but", "if"), etc. As our algorithm solely requires a forward pass and no training, this enhances computational efficiency. To further optimize performance, we employ an Nvidia RTX A5000 GPU. All experiments are conducted on an Ubuntu server equipped with an AMD Ryzen Threadripper 3990X 64-Core Processor and 256 GB of RAM. The job post data is publicly available at `https://www.kaggle.com/datasets/sachinmeena04/linkdin-jobs-dataset`.

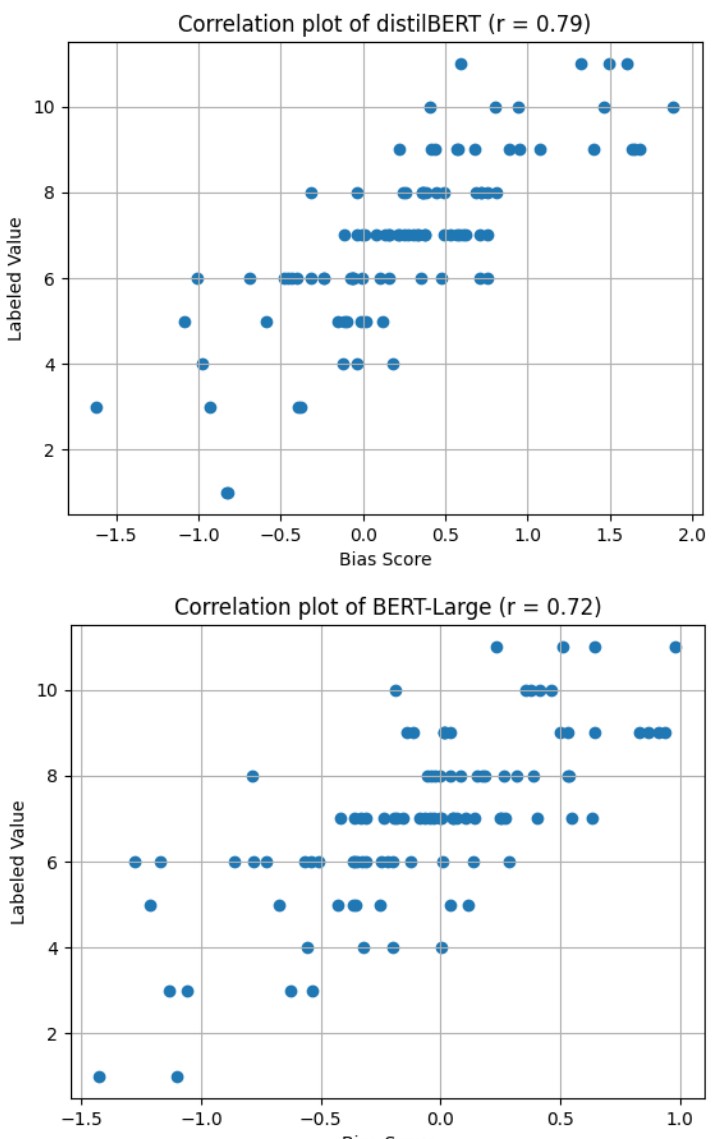

Figure 8: Human Label Evaluation Correlation

### A.9 Word List Details

The full word list is the integration of the existing literature in sociology, we directly adopted them without modification. These word lists are already publicly available. Below is the table for the size of each word list:

Table 6: Counts of gendered words in categories

|      | Fem | Mas |
| --- | --- | --- |
| psy  | 73  | 120 |
| role | 7   | 53  |
| wfc  | 28  | 25  |
| gsc  | 6   | 6   |

As well as some job posting examples:

Table 7: Examples of Gendered Lexicon and Job Advertisement Excerpts

| Gender | Example Lexicon | Example Job Advertisement Excerpts |
| --- | --- | --- |
| Masculine | confident, effective, innovative, (pro)active, practical, pragmatic, problem-solving | - **Confident** to work with **high-caliber** people.
- Proven ability to be **effective** in a fast-paced, ambiguous, and changing environment.
- Encourage new ideas and **innovative** approaches and **actively** share knowledge and experience to enhance the development of the team. |
| Feminine | attentive, accurate, timely, caring, polite, diplomatic | - You have strong **attention to detail**.
- Responsible for the **timely** and **accurate** maintenance of accounting systems at [town name].
- High level of initiative, maturity, **tact** and **diplomacy**. |

#### A.9.1 Limitations, Discussions, and Future Works

**Tokenizer Issue:** Discrepancies arise from the adaptation of word lists for use in MLMs due primarily to tokenization strategies. For example, the word "limitless" is tokenized into ['limit', '##less'], which may affect the representation and subsequent analysis of such words in studies of linguistic bias.

**Bias in MLMs:** Addressing inherent social biases within MLMs presents a significant challenge and offers fruitful directions for future research:

- A primary consideration is our objective to evaluate bias in LLMs. Specifically, we examine the **change in the bias score** between the input and output text. By using **the same MLM** for both input and output, we can isolate and measure any additional bias introduced during the text generation process. This approach also helps control for the baseline biases present in the MLM being used for evaluation.

- A more fundamental question is whether a biased model is essential for picking up the biased words to be able to detect bias. If so, arguably, the original BERT model, trained on extensive real-world data, effectively mirrors societal biases, thus providing a realistic framework for identifying and analyzing these biases. But we still don't have a clear answer to this question.

- In practice, achieving a perfectly unbiased model is currently beyond our reach. The pursuit of such an ideal model remains theoretical at best, as even the debiased MLMs carry traces of inherent biases from their training corpora.

All our results are based on an English dataset; however, additional complexities may arise with other languages due to more intricate word splitting or tokenization challenges. The Masked Language

Models (MLMs) utilized in our study are sourced from public, open-source pre-trained models. The inherent biases of these pre-trained models might impact our results, although we have attempted to mitigate this issue through a robust rank-based method that reduces sensitivity to changes in probability distributions. Currently, our job application generation relies on basic prompts; exploring the effects of varied prompts to capture a broader spectrum of biases constitutes part of our future work. Moreover, while our framework is initially designed to use pairs of word lists, it possesses the flexibility to accommodate single or multiple word lists with minimal adjustments, an extension we also plan to explore in future research endeavors.

