# OpenReview forum: "Probing Social Bias in Labor Market Text Generation by ChatGPT: A Masked Language Model Approach"
_NeurIPS.cc/2024/Conference — NeurIPS 2024 poster_

### Official Review · Reviewer_HqWb · 2024-07-04

**Soundness:** 2
**Presentation:** 4
**Contribution:** 2
**Rating:** 3
**Confidence:** 4

**Summary:**

This paper presents a method to assess the extent to which social biases are encoded in LLMs. The motivating example throughout the paper is assessing the degree of gender bias in the context of job applications in the labor market. The method (PRISM) is designed as follows: given text (presumably output by the LLM), each word is masked out, one at a time, and the masked word is predicted using a masked language model. However, the only masked words predicted belong to one of two lists, corresponding to the two social groups (e.g. a list of masculine terms and a list of feminine terms). If the probability of words in one list are consistently larger than that of the other list, there is evidence for social bias.

The paper then uses PRISM to assess the social biases of chatGPT. chatGPT is prompted to produce job applications, and the method analyzes bias across four social dimensions relating to gender. The paper finds that biases in job postings are likely reproduced and even exacerbated in job applications generated by chatGPT.

**Strengths:**

Strengths:

- Important problem: the paper addresses the important problem of detecting and understanding social biases. The increasing adoption of LLMs by both employers and job seekers makes this a relevant issue. The potential of these models to amplify biases underscores the importance of studying this problem.

- Intuitive method: PRISM is an intuitive method that's easy to understand and explain. Moreover, it's flexible and efficient. All that's needed is word lists and an MLM to use for evaluation.

- Writing: The writing is clear throughout the paper, which helps make the method easy to understand.

**Weaknesses:**

The biggest weakness of the paper is the lack of evaluation. A new method is proposed for assessing social bias in LLMs, but only a small portion of the paper is dedicated to validating the new algorithm. While the validation exercises are a good start, they're limited; for example, the human experts validation is just limited to one dataset, with 6 human labelers and <100 examples. To see evidence that this is a general method, we need to see robustness across multiple application areas and experiments.

Related to the above, I can imagine a problem of this approach where biases of the masked language model are instead incorrectly attributed to the LLM generating the text. For example, if the masked language model used is biased (e.g. is likelier to predict male words than female words), then any sequence of text PRISM is evaluated on will be viewed as biased. It seems that for this test to be reliable, the masked language model should form calibrated and "unbiased" predictions of its own. Couldn't this be revealing biases in BERT rather than the LLM?  I'm not sure how likely this is, but it warrants discussion and validation in the paper; otherwise the limited evaluation is not compelling.

An additional weakness is that the experiments are limited. The experiments don't include baselines for assessing social bias. Is there evidence that PRISM is picking up on biases that other methods miss? Additionally, only one LLM is considered in the job posting experiment, and one MLM (BERT) is used as the MLM -- ablating the latter could help demonstrate the robustness of the method.

A little less urgent, it would've been helpful to see more qualitative examples of the postings, along with more details for the experiments (e.g. the full word lists or at least the lengths of each word list).

**Questions:**

See section above, namely:
- Is there evidence that the test isn't susceptible to social biases in masked language models?
- What does PRISM find that isn't found by other methods?
- Is there more evaluation/validation that the method is useful?

**Limitations:**

Yes

---

> ### Author Rebuttal · Authors · 2024-08-06
>
> Thanks for the detailed and thorough comments. Below please find our responses to specific comments:
>
> ---
>
> W1:  We acknowledge and face challenges in evaluating social bias in text:
> 1. The evaluation benchmarks in this field are scarce and underdeveloped so we can only turn to human labeling. This is also the key motivation for developing our method, which aims to provide a reference bias score for a field lacking domain-specific labeled data.
> 1. Compared to related papers in this field, our evaluation is more robust. Many existing evaluations only rely on heuristics, such as counting gendered words like “he/she” [3,8].
> 1. Evaluating gender bias is particularly challenging due to its subtle and multifaceted nature. This complexity makes it difficult to use crowdsourcing effectively, leading us to rely on expert labeling. Our paper significantly advances existing research by providing expert validation across multiple dimensions of gendered language in the labor market context. While not perfect, our approach represents the first of its kind to provide such comprehensive expert validation, addressing a critical gap in the literature.
>
> Specifically,  we have used the following strategies for validation:
> 1. For human evaluation, we work with experts in related fields (sociology, psychology, management, human resources), with an iterative multi-round coding process.  This ensures high labeling quality, especially compared to non-expert crowdsourcing workers (who may be less attuned to the intricacies of labor market dynamics). We have provided more details about the expert labeling process in the global response.
> 2. In addition to human evaluation, we also conducted benchmark validation. The task has more than **250,000** examples and the dataset is from a different domain about classifying biography.
> 3. Bias is inherently contextual; what is biased in one context may not be in another. To ensure accuracy and relevance, we focused specifically on the labor market, going beyond generic language use that may not apply well to labor market processes.
>
> Finally, the main goal of this paper is to evaluate the bias within LLMs. Please refer to our response to your next question for additional details.
>
> ---
>
> W2: This is a really good and difficult question.
>
> First, we want to emphasize that our final goal is to assess LLM bias, i.e., we are investigating the **change in the score** between input text and output text. By using **the same MLM** for input text and output text, which allow us to isolate and measure the additional bias introduced by the LLM during the text generation process, thereby controlling for the baseline biases present in the MLM used for evaluation.
>
> Additionally, we conducted **Control Experiments with Different MLMs**, we selected two more MLMs: BERT-large and DistilBERT. The results for both the human validation score and the LLM correlation plots are provided in the additional PDF. These results demonstrate that changing the MLMs produces consistent outcomes in the scatter density plot for evaluating bias for LLM, and statistical testing results are also highly consistent. Also in lines 507-510, the use of the ranking method helps mitigate bias within MLMs.
>
> Indeed, the bias for each MLM affects the bias score calculation, but we would like to highlight that related research, such as using word embeddings for bias score[3], also faces similar questions regarding the bias inherent in the embeddings themselves. This is a big challenge in the field, but we do our best to conduct control experiments and validation methods to mitigate this issue.
>
>
> ---
>
> W3: We include baselines in the benchmark validation (see Lines 210-218 and 482-490). One baseline uses pure counting while the other uses static word embedding (word2vec). Our method is the first of its kind that uses contextual information with an MLM, i.e., the information of the whole sentence, other than looking at each word individually. From Figure 4b in the paper, we can observe a better performance compared with other baselines.
>
> ---
>
> W4:  The full word list is the integration of the existing literature in sociology, specifically references[5,6,7] and we directly adopted them without modification. These word lists are already publicly available. And we will publish the wordlists and the corresponding code.
>
> Below is the table for the length of each word list:
> | | Fem | Mas |
> |------|-----|-----|
> | psy| 73  | 120 |
> | role | 7   | 53  |
> | wfc  | 28  | 25  |
> | gsc  | 6   | 6   |
>
> As well as some job posting examples:
>
> | Gender     | Example lexicon                                               | Example job advertisement excerpts                                                                 |
> |------------|---------------------------------------------------------|------------------------------------------------------------------------------------|
> | Masculine  | confident, effective, innovative, (pro)active, practical, pragmatic, problem-solving | - **Confident** to work with **high-caliber** people.  - Proven ability to be **effective** in a fast-paced, ambiguous, and changing environment. - Encourage new ideas and **innovative** approaches and **actively** share knowledge and experience to enhance the development of the team. |
> | Feminine   | attentive, accurate, timely, caring, polite, diplomatic | - You have strong **attention to detail**. - Responsible for the **timely** and **accurate** maintenance of accounting systems at [town name]. - High level of initiative, maturity, **tact** and **diplomacy**. |

---

> > ### Comment · Reviewer_HqWb · 2024-08-09
> >
> > I thank the authors for responding to my comments. I will maintain my original score

---

> > > ### Author Response · Authors · 2024-08-09
> > >
> > > Dear reviewer HqWb, thank you for your reply. We wonder if there is any further information/clarification we can provide to help clarify your remaining queries and bolster your confidence in our work. Thank you.

---

### Official Review · Reviewer_ZJXv · 2024-07-08

**Soundness:** 3
**Presentation:** 3
**Contribution:** 2
**Rating:** 5
**Confidence:** 3

**Summary:**

This research investigates the social biases present in ChatGPT-generated job applications using a novel experimental design that responds to real job advertisements. By simulating the job application creation process, the study examines the language patterns and biases that surface when the model is prompted with diverse job postings. Specifically, this paper presents a PRISM algorithm based on Masked Language Models to quantitatively assess social bias based on validated inventories of social cues/words. The results show that the increasing adoption of generative AI can reinforce and exacerbate gender and social inequalities in the labor market through the use of biased and gendered language.

**Strengths:**

1. This paper provides a comprehensive analysis of the bias problem in LLMs, clearly presenting the motivations behind the model design.

2. A key contribution of this research is the development of a novel bias evaluation framework based on Masked Language Models, which quantitatively assesses social bias using validated inventories of social cues and words. This framework advances existing methods in terms of efficiency and flexibility, requiring minimal human labeling efforts.

3. The model evaluation is convincing, featuring a well-designed experiment to probe the black-box of social biases in ChatGPT models.

**Weaknesses:**

1. The bias evaluation in this paper highly relies on the predefined bias-related word lists, e.g., the Feminine word list and Masculine word list, so how the quality of the word lists will affect the bias evaluation results? It would be better to make analysis on this aspect.

2. The proposed method appears to be generally applied to any domain, although the major focus is on labor market text generation. Therefore, the characteristics of bias in labor market text generation need to be highlighted.

3. There are some writing problems in this paper, for example, the table captions should generally be presented above the table.

**Questions:**

1. How the quality of the word lists will affect the bias evaluation results?

2. The co-occurring frequency of some words are naturally generated, for example, in our common understanding, soft is more related to woman instead of man. Is it reasonable to treat it as bias?

**Limitations:**

The major limitation is the validation of the basic motivation of assessing social bias based on validated inventories of social cues/words.

---

> ### Author Rebuttal · Authors · 2024-08-06
>
> We would like to express our sincere gratitude to the reviewer for the time and effort in providing the comments and suggestions. Below, we address each of your points in detail.
>
> ---
>
> W1: Thank you for your question, which we address in three aspects:
> 1. **Use Case**: In our settings, we used pre-validated word lists that are widely used in existing research. Our method aims to provide a biased score based on words chosen by these lists that are grounded in expert knowledge of labor market processes across the disciplines of sociology, management/human resources, psychology, and linguistics. Therefore, we are confident in the validity and reliability of the widely used and social science expert-validated word lists.
> 2. **Algorithmic Aspect**: The design of our algorithm incorporates robustness measures. Specifically, after combining the two sets of probabilities, we only take the top alpha percentage for calculation.
> 3. **Numerical Experiment**: We have conducted additional experiments to assess the impact of the word-list quality. As shown in Figure 4(b) in the new PDF, the correlation with human labeling decreases as we remove more words from the word list. This additional analysis helps underscore the importance of a comprehensive and accurate word list for reliable bias evaluation results.
>
> ---
>
> W2: This is an important point. We want to first explain that biases are contextual, i.e. some words are biased in one domain but not in another. So in order to be correct and precise, the word lists we use are grounded in multidisciplinary literature on labor market processes, which goes beyond generic language use that may not necessarily apply well to the labor market context. Our method is easily generalized to other domains by changing the word lists (to other domain-specific ones). Moreover, drawing on interdisciplinary collaboration, we believe our work is a good illustration of combining ML and social science, and we believe our paper fits well into the Neurips stream of ‘Machine learning for social sciences’.
>
> ---
>
> W3: Thank you for pointing this out. We have fixed the issue you raised and moved the table titles so that they appear above the tables.
>
> ---
> Q1:  same as W1
>
> ---
> Q2:  In terms of the specific example you noted, established psychological methods and evidence shows that language associated with characteristics such as “soft” is perceived by readers as “feminine”, which tends to appeal to and resonate more with women rather than men. In the case of job posting and application, this would mean that (1) the use of feminine language (e.g., soft) in job postings tends to appeal more to women (than men) applicants; (2) the use of feminine language in job applications tends to be evaluated more favorably for a post stereotypically perceived to a feminine orientation. Therefore, our definition of bias is embedded in the broader social science and psychological literature on the “biased” impact of language use. We have clarified our approach to bias in our revised paper.
>
> This question also pertains to the validity of the word list we use, which we clarify in three aspects:
>
> 1. The word lists are established, widely used, and validated in the literature, specifically those developed by [5,6,7]. They are theoretically grounded in social science research.
> 2. The word lists are specially informed by extensive sociological, psychological, and management research focusing on gender in the labor market.
> 3. Lastly, we conduct a rigorous internal validation process, achieving a high level of intercoder reliability with experts specializing in sociology, management, and labor market research.
>
>
>
> ---
>
> Limitation: We appreciate that some may view this as a limitation, but we view this as a strength. A great deal of social science research in psychology, linguistics, and sociology has established the relationship between language and social bias. We are able to draw on the strengths of decades of research to inform innovations in ML/NLP. Therefore, we believe our paper fits well into the Neurips stream of ‘Machine learning for social sciences’ by illustrating the strengths of interdisciplinary cross-fertilization.

---

### Official Review · Reviewer_Bg7r · 2024-07-11

**Soundness:** 3
**Presentation:** 4
**Contribution:** 3
**Rating:** 8
**Confidence:** 4

**Summary:**

This paper investigates the impact of social biases in the application of LLMs within the labor market. The authors examine biases in job applications generated by ChatGPT based on given job posts. They introduce a new bias evaluation method called PRISM (Probability Ranking bIas Score via Masked language model), which combines predefined gender-related word lists from social sciences with a masked language model approach to assess biases in input texts. Experiments with ChatGPT revealed that biases present in job postings are not only replicated but also amplified in job applications created by generative AI in response to these postings.

**Strengths:**

1. This paper is well written. The authors have clearly made their scopes and the reading of the paper is very comfortable.
2. This topic is novel and interesting and could serve as a good study for ML in social sciences.

**Weaknesses:**

1. Experiments are limited. Only ChatGPT has been applied.
2. This paper aims to investigate the social biases, but no detailed exploration in different bias types has been conducted. From the algorithm, I only see the gender bias involved, which I think is only a part of the social biases.

**Questions:**

1. Based on the scoring function in line between 155 and 156, why do you state "A positive score indicates a bias toward a masculine orientation, while a negative score suggests a bias toward a feminine orientation."? Shouldn't it be the reversed reading?
2. As the word lists for bias evaluation are predefined, how do you confirm their reliability?
3. I am interested in the human evaluation part of sec 3.4. Could you include more details about the human experts and the tasks they have done maybe in the appendix, i.e. the demographics, the inter-annotator agreements, and the amount of tasks they did?

**Limitations:**

Same as the points in Weaknesses

---

> ### Author Rebuttal · Authors · 2024-08-06
>
> We are very grateful for the thoughtful comments and heartwarming acknowledgment, as well as the time and effort devoted to the reviewing process. Below please see our response to the specific point in the reviewing comment.
>
> ---
> W1: Our experiments were specifically conducted using ChatGPT due to its wide accessibility and prevalent usage. This choice was deliberate to ensure that our findings are representative of the typical user experience with one of the most commonly used models. However, we acknowledge the importance of testing additional models, and we agree that future work should include experiments with a broader range of models to validate and extend our findings.
>
> ---
>
> W2: This is a great suggestion, pointing out an important direction for future research, which we have now acknowledged in our revised paper. We focused on gender, as gender inequality in the labor market is well-documented, persistent, and associated with gendered language, making this a prominent and exemplary type of social bias. We were also motivated in part to include just one aspect of bias due to the limited space of the paper. Adding additional dimensions would require providing background on each form of inequality, as well as locating established word lists of linguistic cues associated with those biases. In our revision, we have more clearly articulated that gender biases are chosen as a focal case of a broader range of social biases and that gender bias is important because it underpins one of the most prominent and persisting forms of inequality in the labor market. We have also suggested that future research explore additional forms of bias and labor market inequality
>
> ---
>
> Q1:Thank you for your question. Because we assign lower numerical ranks to higher probabilities. As illustrated in Figure 2 in the paper, a rank of 1 represents the highest probability. Therefore, a lower mean rank indicates higher probabilities, while higher ranks (larger numerical values) correspond to lower probabilities. We have clarified this in our revised paper to ensure the scoring logic is explicitly defined.
>
> ---
>
> Q2: The validity of our word lists is grounded in in three aspects:
>
> 1. The word lists are established, widely used, and validated in the literature, specifically those developed by [5,6,7]. They are theoretically grounded in social science research.
> 2. The word lists are specially informed by extensive sociological, psychological, and management research focusing on gender in the labor market.
> 3. Lastly, we conduct a rigorous internal validation process, achieving a high level of intercoder reliability with experts specializing in sociology, management, and labor market research.:
>
> ---
>
> Q3: Thank you for pointing this out. We will include this additional information in the appendix:
>
> To ensure the scientific rigor of the evaluation, we paid particular attention to inter-rater validity and reliability. Specifically, each phase included individual labeling of data conducted by four independent experts specializing in labor market inequalities associated with gender, work, and family, and Equity, Diversity, and Inclusion (EDI) in the labor market. In each round of labeling, individual labeling was followed by group sharing and discussion of the preliminary outcomes among the four experts. This combination of individual and group analysis allowed the team to trace the score labeling and the validation of word inventory, ensure inter-rater reliability in each phase, and contextualize each phase within relevant scholarly literature, policies, and definitions.
>
> The score labeling and the validation were further validated by three additional expert labelers (management, human resource, and social science scholars) from the team. Through a double-blind labeling approach, the three additional experts independently assessed the dimensions and labels produced by the first four experts, demonstrating a high level of consistency. The scores and word lists were then finalized through further deliberation among the four experts and the three additional validators.
>
> Because we used an iterative multi-round coding process, the inter-coder consistency rate in the developmental coding varied between 0.6 and 0.8. Notably, the final validation by three fresh validators within the team achieved a high level of inter-coder consistency exceeding 0.8.

---

> > ### Comment · Reviewer_Bg7r · 2024-08-14
> > **Thank you**
> >
> > Thank you for the response. I think my initial questions were properly addressed. I will raise my score and would be glad if this paper could be included to the proceedings, which I think would definitely diversify the conference.

---

> > > ### Author Response · Authors · 2024-08-14
> > >
> > > Dear Reviewer Bg7r, we truly appreciate your valuable support and recognition of our paper!

---

### Official Review · Reviewer_e7QL · 2024-07-11

**Soundness:** 1
**Presentation:** 3
**Contribution:** 2
**Rating:** 7
**Confidence:** 4

**Summary:**

This paper presents a novel bias detection algorithm, PRISM, and uses it to evaluate biases in job applications generated from prompts that include job postings. The technique uses a masked language model (MLM) to find the probability of “masculine” and “feminine” words replacing the masked word in the text. The bias score is then the difference between average ranks of feminine and masculine words, averaged over all words in the text. The authors find a correlation between biased job postings and biased generated text based on their evaluation method.

**Strengths:**

This paper’s primary strengths are that the work is well presented and the text is clear. The problem being tackled, gender bias in LLMs, is also one that is critically important and deserving of study.

**Weaknesses:**

I have two significant concerns about this work. First, I have a meta-concern about the setting in which the experiments are being conducted. Is generating a job application from a posting a common use case of LLMs? If there is indeed exacerbation of bias going from posting to application, what are the concrete harms of that manifesting itself? In short, I do not see much in the text motivating the work and helping us understand the real world impacts of these issues.

Second, I have a specific concern about the methodology employed. The paper claims that the PRISM method obviates the need for human labeling, and this seems to be accomplished through the use of the MLM and the previously curated lists of words. I am not convinced based on what I have seen in the paper that this method is actually measuring the level of bias in the generated text - it may instead be measuring the gender bias in the underlying MLM (bert-base-uncased) that is used to compute the score. It seems there is a fundamental chicken and egg problem. Is a “feminine” word ranked higher because the surrounding text that was generated is biased, or is it ranked higher because the MLM is biased to predict more feminine words in that context when a neutral word would actually be perfectly appropriate. How would the authors design control experiments to disentangle these two effects?

**Questions:**

Meta question: isn’t the assumption that certain words are “feminine” or “masculine” already a sort of gender bias? I suspect that the paper means that these buckets are words that are stereotypically or historically associated with certain genders, but if that is the case it should be made explicit.

Can the authors publish the exact word lists used for the experiment?

Why are means of ranks employed as opposed to the means of the probabilities themselves?

Similarly, is averaging the S scores to compute B(T) the right approach? What do the actual distributions of S look like? Is the mean capturing the distribution faithfully?

What version of ChatGPT was used for this experiment? Because the models are updated fairly frequently, just specifying the use of the API is not sufficient for reproducibility.

**Limitations:**

The paper discusses limitations briefly in the appendix, and there is a very brief discussion about the potential for biases in the MLM to affect the results. However, I believe there needs to be significantly more validation that this approach is sound based on the concerns I enumerated above.

---

> ### Author Rebuttal · Authors · 2024-08-06
>
> Thanks for the detailed and thorough comments. Below please find our responses to specific comments:
>
> ----
>
> W1: the use of LLMs like ChatGPT for generating job applications is indeed increasingly common among job seekers. This trend has been highlighted in discussions and concerns raised by employers and HR professionals, as reported by high-profile news such as CNBC (May 6, 2024, article: "Exact same cover letters word for word: Career consultant says Gen Z are misusing AI") and Vox (Mar 8, 2023, article: "Maybe AI can finally kill the cover letter: Jobs still require cover letters. Apps like ChatGPT can help."). Moreover, the academic community is beginning to explore this use case, focusing both on its technical applications[1] and ethical/bias implications[2]. Our study contributes to this burgeoning body of literature by addressing key public, organizational, and scholarly concerns regarding the use of LLMs in generating job applications.
>
> We explicitly identify the two major harms resulting from bias exacerbation in LLM-generated job applications:
>
> 1. **Structural Harm**: LLM-generated job applications that perpetuate gender stereotypes contribute to the reinforcement of gender inequalities embedded in language used in labor market processes. Job postings and applications are critical steps in these processes and play a significant role in the reproduction of gender inequalities. By failing to challenge these biases, LLMs can inadvertently support the perpetuation of these inequities.
>
> 2. **Practical Harm**: As gender-biased LLM-generated job applications become part of the training data for future AI applications in HR(both in generating job postings and assessing applications), there is a risk of further entrenching gender biases in language use. This entrenchment can lead to cascading effects, such as increased labor force gender segregation, which have significant societal implications.
>
> ---
>
> W2: This is a really good question. We want to emphasize that our goal is to assess LLM bias, i.e., we are investigating the **change in the score** between input and output text. Using the **same MLM** for input and output text, allows us to isolate and measure the additional bias introduced by the LLM during the text generation process, thereby controlling for the baseline biases present in the MLM used for evaluation.
>
> Additionally, we conducted **Control Experiments with Different MLMs**, we selected two more MLMs: BERT-large and DistilBERT. The results for both the human validation and the LLM correlation plots are in the additional PDF. These results show that changing the MLMs produces consistent outcomes in the scatter density plot for evaluating bias for LLM, and statistical testing results are also highly consistent.
>
> Indeed, the bias for each MLM affects the bias score calculation, but we would like to highlight that related research, such as using word embeddings for bias score[3], also faces similar questions regarding the bias inherent in the embeddings themselves. This is a big challenge in the field, but we do our best to conduct control experiments and validation methods to mitigate this issue.
>
> ---
>
> Q1: You are correct about our intended meaning, that words are stereotypically or historically associated with certain genders. We clarified that our word lists capture feminine and masculine orientations that are grounded in and associated with gender norms and stereotypes, as supported by extensive sociological, psychological, and linguistics research[4,5,6]. Notably, while some of this work is more historical in nature(evidence from decades ago), more recent research cited below also establishes the continuing importance of gendered language in contemporary settings.
>
> ---
>
> Q2: We would like to clarify that the word lists used in our experiment were not created by us specifically for this paper. Instead, we directly adopted them from existing literature across the social science research as referenced in [5,6,7], without modification. These word lists are publicly available, and we will publish the word lists along with the corresponding code.
>
> Using pre-validated word lists is a key advantage of our method, allowing us to easily incorporate established, rigorous, and theoretically grounded word inventories from social science research (see lines 132–136).
>
> Additionally, this paper also draws on the expertise of several co-authors specializing in sociology, management, and labor market research to provide further internal validation of the word lists.
>
> ---
>
> Q3: Rank has several advantages over probabilities:
>
> **Theoretical Properties**: Ranks exhibit normality and allow for a rigorous formulation of the test statistic and its asymptotic result as presented in Theorem 1, which is based on the Wilcoxon rank sum test. Probabilities do not possess these important properties.
>
> **Sensitivity and Robustness**: Direct use of probabilities can be overly sensitive due to the highly skewed nature of prediction distributions. By using ranks, we can mitigate this sensitivity, and enhance the robustness of our results. This robustness enhancement also helps mitigate the inherent biases present in these pre-trained models.
>
> ---
>
> Q4: From a Bayesian view, when we lack prior information, the mean serves as a robust estimator of central tendency. However, we acknowledge that for future work, incorporating prior information for a weighted average or other aggregation methods could potentially improve performance.
>
> The asymptotic distribution of S is normal. To validate this, we have included a histogram of S in the PDF. Additionally, we conducted the Shapiro-Wilk test, which resulted in a p-value of 1.00, confirming that S is indeed normally distributed
>
> ---
>
> Q5: We employed the GPT-3.5 Turbo, as it was the most accessible and widely used free version at the time of our experiment. This choice was made to ensure that our results are representative of the typical user experience.

---

> > ### Comment · Reviewer_e7QL · 2024-08-08
> >
> > Thank you to the authors for their thorough responses. I believe you have addressed my concerns about the weakness of the MLM usage. As such, I am adjusting my score to 7.

---

> > > ### Author Response · Authors · 2024-08-09
> > > **Thank you**
> > >
> > > Dear reviewer e7QL, thank you for your kind support and recognition of our work.

---

### Author Rebuttal · Authors · 2024-08-07

Dear reviewers,

Thank you so much for your time and effort in reviewing our paper. We want to highlight some key aspects of our paper:
1. Our method represents the first research on unsupervised bias evaluation in text using contextual information.
2. Our word lists are based on established word inventories, grounded in social science/labor market theories and tried and tested in a wide range of research (cited below). Our team, particularly social science scholars in our team, have provided further expert validation of the word lists to doubly ensure the validity and robustness of the word lists.
3. Our work exemplifies the interdisciplinary synergy between machine learning and social science, aligning well with the NeurIPS stream of "Machine Learning for Social Sciences." We hope the reviewers will consider the significant social science impact and contributions of our paper.



---

Our team's human validation detail:

To ensure the scientific rigor of the evaluation, we paid particular attention to inter-rater validity and reliability. Specifically, each phase included individual labeling of data conducted by four independent experts specializing in labor market inequalities associated with gender, work, and family, and Equity, Diversity, and Inclusion (EDI) in the labor market. In each round of labeling, individual labeling was followed by group sharing and discussion of the preliminary outcomes among the four experts. This combination of individual and group analysis allowed the team to trace the score labeling and the validation of word inventory, ensure inter-rater reliability in each phase, and contextualize each phase within relevant scholarly literature, policies, and definitions.

The score labeling and the validation were further validated by three additional expert labelers (management, human resource, and social science scholars) from the team. Through a double-blind labeling approach, the three additional experts independently assessed the dimensions and labels produced by the first four experts, demonstrating a high level of consistency. The scores and word lists were then finalized through further deliberation among the four experts and the three additional validators.

Because we used an iterative multi-round coding process, the inter-coder consistency rate in the developmental coding varied between 0.6 and 0.8. Notably, the final validation by three fresh validators within the team achieved a high level of inter-coder consistency exceeding 0.8.

---

Reference:

[1] Al Shalabi, H., Al-Hashemi, R., & Al-Ramadin, T. A. (2013). Automatic Cover Letter Generator System from CVs (ACLGS). Global Journal of Computer Science and Technology Software & Data Engineering, 13(3).

[2] Bárány, A. (2023). From Humans to Machines: Can Artificial Intelligence Outsmart Human Job Applications?.

[3] Dhamala, J., Sun, T., Kumar, V., Krishna, S., Pruksachatkun, Y., Chang, K. W., & Gupta, R. (2021, March). Bold: Dataset and metrics for measuring biases in open-ended language generation. In Proceedings of the 2021 ACM conference on fairness, accountability, and transparency (pp. 862-872).

[4] Coates, J. (2015). Women, men, and language: A sociolinguistic account of gender differences in language. Routledge.

[5] Bem, S. L. (1974). The measurement of psychological androgyny. Journal of consulting and clinical psychology, 42(2), 155.

[6] Gaucher, D., Friesen, J., & Kay, A. C. (2011). Evidence that gendered wording in job advertisements exists and sustains gender inequality. Journal of personality and social psychology, 101(1), 109.

[7] Konnikov, A., Denier, N., Hu, Y., Hughes, K. D., Alshehabi Al-Ani, J., Ding, L., ... & Tarafdar, M. (2022). BIAS Word inventory for work and employment diversity,(in) equality and inclusivity (Version 1.0). SocArXiv.

[8] Wang, R., Cheng, P., & Henao, R. (2023, April). Toward fairness in text generation via mutual information minimization based on importance sampling. In International Conference on Artificial Intelligence and Statistics (pp. 4473-4485). PMLR.

---

### Decision · Program_Chairs · 2024-09-25

**Decision:**

Accept (poster)

**Comment:**

The review team mostly agrees that the paper is well-written and studies a novel topic within the emerging area of ML for the Social Sciences.

There are some concerns about whether the evaluation process is robust and appropriate for the problem. During the reviewers’ discussion after the rebuttal, the main concern was that the paper used an MLM to measure the bias of another LLM. The argument was that if the MLM is biased then it's possible that biases of the MLM are attributed to the LLM. Although a counter-argument was that measuring the change between the input text and output text partially addresses this issue, it doesn’t completely eliminate the concern.

Another secondary concern was that the human experts validation is just limited to one dataset, with 6 human labelers and <100 examples. In their rebuttal, the authors responded with several quite convincing arguments including that for human evaluation, they worked with experts in related fields (sociology, psychology, management, human resources), with an iterative multi-round coding process and that in addition to human evaluation, they also conducted benchmark validation with more than 250,000 examples and a dataset from a different domain. Illustrating these arguments in the paper would be helpful.

The review team pointed out several suggestions for revisions.

- Address the concern about using an MLM to measure the bias of another LLM. (See Review HqWb)
- Include more details about the human experts and the tasks they have done maybe in the appendix, i.e. the demographics, the inter-annotator agreements, and the amount of tasks they did. (See Review Bg7r)
- Include the predefined bias-related word lists (or at least details about) and better argue how the quality of the word lists may/ may not affect the bias evaluation results (see Review ZJXv)

Based on the information the authors provided in the rebuttal, I believe that at least the last two concerns are easily addressable. Overall, the strengths of the paper generally outweigh the aforementioned concerns.